# Preparation of Core-Shell-Structured RDX@PVDF Microspheres with Improved Thermal Stability and Decreased Mechanical Sensitivity

**DOI:** 10.3390/polym14204262

**Published:** 2022-10-11

**Authors:** Hulin Wu, Aifeng Jiang, Mengru Li, Yanyan Wang, Fangchao Zhao, Yanchun Li

**Affiliations:** 1Southwest Technology and Engineering Research Institute, Chongqing 400039, China; 2Bishan Joint Research Institute of Advanced Technology, Chongqing University, Chongqing 400044, China; 3School of Chemistry and Chemical Engineering, Nanjing University of Science and Technology, Nanjing 210094, China

**Keywords:** RDX, core-shell-structure microspheres, thermal stability, friction sensitivity, impact sensitivity

## Abstract

Reducing the sensitivity of high-energy simple explosives is the key technology in improving the practical application of high-energy insensitive powder. As the most widely used high-energy explosive, hexahydro-1,3,5-trinitro-1,3,5-triazine (RDX) is limited in application due to its high sensitivity. In this work, polyvinylidene fluoride (PVDF) was used as an energetic binder. Core-shell-structured RDX@PVDF microspheres are produced using electrospray assembly technology and fully characterized by thermogravimetric analysis, X-ray diffraction, scanning electron microscopy, transmission electron microscopy, energy dispersive spectroscopy, and mechanical sensitivity. Their thermal stability and mechanical sensitivity are directly related to the weight fraction of the added PVDF. Moreover, core-shell-structured RDX@PVDF microspheres with RDX and PVDF in the proportion three to one possess a spherical-like morphology, the lowest impact sensitivity, the lowest friction sensitivity, and the highest thermal stability. This work provides a facile method for the positive design energetic materials and the prediction of their environmental adaptability.

## 1. Introduction

Hexahydro-1,3,5-trinitro-1,3,5-triazine (RDX), a high-energy nitroamine explosive with strong explosive power and high-energy density, has been widely used in casting explosives and high-energy propellants [1,2,3]. Currently, industrial RDX shows a coarse surface, considerable internal defects, and poor dispersion [4,5], which reduces charging density and increases sensitivity during practical applications. To meet the requirements of high energy, low sensitivity, and high environmental adaptability, various modified methods, including refinement [6], coating particles with binder [7], and spheroidizing [8], have been applied to effectively reduce the sensitivity of nitroamine explosive. Coating treatment is widely utilized to modify RDX, due to its advantages of controlling the agglomeration of particles, improving dispersion, enhancing combustion characteristics, and realizing multi-functional composite particles [9,10].

Recently, the main coating materials employed to reduce sensitivity of RDX have included Viton (dipolymers of hexafluoropropylene and vinylidene fluoride) [9], DAAF (3,3′-diamino-4,4′-azoxyfurazan) [10], F2604 (FE2604-2, a kind of fluoro-rubber) [11], PMMA (polymethyl methacrylate) [12], MUF (melamine-urea-formaldehyde) [13], and NC [14,15,16,17]. Jia et al. [14] fabricated honeycomb-like structured nanoparticles in which partial RDX crystals were encapsulated in PMMA-PVA (polymethylmethacrylate-polyvinyl acetate) coating, manifested that PMMA could effectively improve the thermal performance of RDX. Yao et al. [11] successfully prepared RDX/F2604 composite particles by electrospray assembly technology, which possess lower impact sensitivity and friction sensitivity compared with that of raw RDX. In order to increase the heat release of composite particles, aluminum particles are also added to composite particles while covering [8,18]. Although much progress has been achieved in the field of RDX coating, there are still many challenges, such as the uniformity of the coating layer, the firmness of the coating layer, adjustments to the morphology of particles while coating, and the reduction in sensitivity without sacrificing energy density.

Recently, polyvinylidene fluoride (PVDF) has attracted researchers’ attention in the application as a binder in composite particles. It is used to coat aluminum powder [19], B/KNO_3_ [20], Al/Fe_2_O_3_ [21], Al/CuO/RDX [22], etc. Herein, PVDF was selected to be the coating layer because of its characteristics of corrosion resistance, high temperature resistance, oxidation resistance, and a strong hydrogen bond [23]. At the same time, fluorine has strong electronegativity; it has been confirmed that fluoride can promote the combustion and explosion of a system [24].

In addition to the coating materials, the coating process also directly affects the properties of composite particles. The common processes for RDX coating include the solution suspension method [25], the water suspension method [26], the lotion polymerization method [27], the self-assembly method [28], physical vapor deposition (PVD) [29], and supercritical solution rapid expansion (RESS) technology [30]. In recent years, electrostatic spray technology has been widely used in the preparation of composite particles [31,32], because it can easily realize the regulation of particle size from tens of nanometers to tens of microns, and it can also control the surface morphology of composite particles [33].

In this study, to improve the thermal stability and reduce the mechanical sensitivity of RDX, PVDF was used as a coating-material to prepare core-shell structured RDX@PVDF composite microspheres using electrospray technology. The morphology performance was characterized by scanning electron microscopy (SEM) and transmission electron microscopy (TEM); element distribution was also measured via energy dispersive spectrometry (EDS). The thermal sensitivity was analyzed by thermogravimetric and differential scanning calorimetry (TGA-DSC) technology. The friction sensitivity and impact sensitivity were also examined by the ⟪GJB 772A-97 explosive test method⟫.

## 2. Experimental

### 2.1. Materials

The materials used in this study were hexogen (Gansu Yinguang Chemical Industrial Group Co., Ltd., Baiyin, China); polyvinylidene difluoride (5 μm, Meryer Chemical Technology Co., Ltd., Shanghai, China); nitrocellulose (1000 of polymerization degree, 11.9% of nitrogenous content, Qingan Chemical Co., Ltd., Xi’an, China); acetone (99.8%, Sinopharm Chemical Reagent Co., Ltd., Shanghai, China); and N,N-dimethylformamide (99.8%, Sinopharm Chemical Reagent Co., Ltd., Shanghai, China).

### 2.2. Preparation of Precursor Solution

The preparation parameters of the precursor solution formulae of RDX@PVDF microspheres are shown in Table 1. Sixty-five mg, 52 mg, and 43 mg of PVDF was weighted and added to the mixed solution of acetone (6 mL) and DMF (2 mL) to fully dissolve. Subsequently, 195 mg, 208 mg, and 217 mg of RDX was added to the abovementioned PVDF solution, which was stirred for 4 h to obtain the precursor solution.

### 2.3. Electrospray of RDX@PVDF Microspheres

The solution of precursor was added to a syringe and flowed out through a metal nozzle (inner diameter 0.51 mm, outer diameter 0.80 mm) with a 1.5 mL∙h^−1^ flow rate, using an injection pump (RSP01-B by Jiashan Ruichuang Electronic Technology Co., Ltd., Jiashan, China). The metal nozzle was linked to a high voltage source (0–30 kV, TE4020, Dalian Taisman Technology Co., Ltd., Dalian, China) with an operation voltage of 23 kV. The aluminum foil used as a receiver was fixed in an insulated plastic plate and grounded. The received distance of the metal nozzle and the aluminum foil was 8 cm. The scheme of electrospray technology is depicted in Figure 1. A high voltage electrostatic field was formed when the voltage was supplied. The precursor solution was atomized to form into a liquid drop under high voltage and surface tension and moved toward the aluminum foil. With the volatilization of the solvent, the microspheres of RDX@PVDF were collected on the aluminum foil. For comparison, the RDX@NC (3:1) composite microspheres were also prepared by the same process.

### 2.4. Morphology Analysis

The sample surface was handled with gold to increase the electrical conductibility. The surface morphology of the samples was characterized by scanning electron microscopy (FEI Nova Nano SEM 450, America, FEI Company, Shanghai, China). Transmission electron microscopy (FEI Talos F200X, America, FEI Company) was used to characterize the coating of the composite particles. The sample element distribution was examined via energy dispersive spectrometry (EDS).

### 2.5. Fourier Transform Infrared Spectroscopy (FTIR) Analysis

FTIR analysis is a universal method in the field of explosives analysis. FTIR spectra were collected by an FTIR spectrometer (NICOLETIS 10, Thermo Fisher Scientific, Shanghai, China). The spectra were recorded in transmission mode in the spectral range of 4000 cm^−1^ to 400 cm^−1^; spectra resolution was maintained at 4 cm^−1^ in all measurements.

### 2.6. Thermal Characterization

The thermal performance of the samples was investigated by Thermogravimetric Analysis (TGA) and Differential Scanning Calorimetry (DSC) (NETZSCH STA449C). After being weighed, the samples were placed in an aluminum crucible (85 μL) with a mass of approximatelhy 1.0 mg, under a high-purity argon environment with a flow rate of 20 mL∙min^−1^ and a temperature range from 30 °C to 300 °C, with a heating rate of 10 °C∙min^−1^.

### 2.7. Characterization of X-ray Diffraction

X-ray diffraction (XRD) patterns of the samples were examined by X-ray diffractometer (D8 Advance, Bruker, Germany with Cu Kα radiation at λ = 1.5418 Å) The scanning rate was from 5° to 60°.

### 2.8. Characterization of Impact and Friction Sensitivity

The tests of impact and friction sensitivity were carried out according to ⟪GJB 772A-97 explosive test method⟫.

The impact explosion probability was measured by BAM BFH12, with the drop hammer of 2 kg at a height 30 cm. The friction explosion probability was measured by BAM Friction Apparatus FSKM-10, with a loading force of 324 N and 360 N.

## 3. Results and Discussion

### 3.1. Composition and Structure of Samples

To obtain the structure of raw RDX and RDX@PVDF microspheres, X-ray diffraction (XRD) was carried out, as shown in Figure 2. The XRD patterns of RDX@PVDF microspheres agreed well with that of raw RDX, suggesting no crystal transition occurred in RDX that was coated with PVDF via electrospray technology. Furthermore, the diffraction peak intensity decreased with the increase of PVDF content, indicating that the crystal size was reduced.

### 3.2. FTIR Spectra Analysis of Samples

The FTIR absorption spectra of raw RDX, PVDF, and RDX@PVDF (3/1) is depicted in Figure 3. For raw RDX (black curve), the absorption peak at 3074 cm^−1^ was ascribable to the −CH stretching vibration; the three absorption peaks, at approximately 1569 cm^−1^, 1528 cm^−1^, and 1386 cm^−1^, corresponded to the stretching vibrations of −NO_2_; another peak at 1264 cm^−1^ was attributed to the nitramine; the peak at 905 cm^−1^ originated from the stretching vibration of the RDX ring. Compared with pure RDX, the peak of the IR spectra of RDX@PVDF had little significant change, and it was difficult to judge the relevant support of its structure. Therefore, the SEM, TEM, and EDS were used to further illustrate the core-shell structure.

### 3.3. Morphology Characterization of Samples

In order to observe the microscopic morphology and to determine the particles’ sizes and the spheroidization degree of the raw RDX and the RDX@PVDF microspheres, SEM was employed as shown in Figure 4 and the ratio of the length to the diameter of the particles was measured (at least 50 particles were measured in each image to obtain the average value). Raw RDX (Figure 4a) was irregularly massive, with a length–diameter ratio of 2.1:1 to 1.53:1, and the average particle size was 12.35 μm. However, when the added mass loading of PVDF was one-fifth of the RDX, the average particles size of RDX/PVDF was reduced to 2.86 μm with the length–diameter ratio of 1.28:1 to 1.01:1 (Figure 4b). Moreover, when the added mass loading of PVDF increased to one-fourth and one-third of RDX, the average particle size of RDX/PVDF gradually reduced to 0.92 μm, with a length–diameter ratio of 1.23:1 to 1:1 (Figure 4c) and 0.6 μm with a length–diameter ratio of 1.19:1 to 1:1 (Figure 4d), respectively. The results indicated that with the increase of PVDF content, the particle size gradually decreased and the particle morphology gradually became spherical, which was in good agreement with the results obtained from XRD patterns. Moreover, when the added mass loading of PVDF was one-third of RDX, the particle size was the smallest, with the highest degree of spheroidization.

To better understand the microscopic structure, transmission electron microscopy (TEM) measurements were conducted (Figure 4d). The TEM image indicated uniform PVDF coats on RDX, with an obvious core-shell structure. The thickness of the coating layer was 0.13 μm to 0.33 μm, with an average value of 0.2 μm.

With the aim of realizing the element distribution of RDX@PVDF microspheres (RDX/PVDF = 3/1), EDS was carried out and the results are shown in Figure 5. The carbon, nitrogen, oxygen, and fluorine elements in the prepared RDX@PVDF microspheres are all displayed clearly, indicating that PVDF is uniformly coated on the surface of the particles. Moreover, the composite particles in Figure 5 show a similar spherical shape, which is in good agreement with the results of the SEM and TEM measurements.

For comparison, SEM and TEM measurements were employed to describe the RDX@NC (3:1) composite microspheres, as shown in Figure 6a,b. The average particle size of RDX@NC microspheres was 0.79 μm and presented an apparent core-shell structure, with a coating thickness of 21 nm to 53 nm and an average thickness of 33 nm. In general, the particle size of NC-coated composite particles was smaller, and the coating layer was thinner than that of PVDF-coated composite particles, which was mainly due to the low viscosity of the NC precursor fluid.

### 3.4. Thermal Stability of Samples

To study the thermal stability of RDX@PVDF microspheres, TGA-DSC measurements were used to examine the decomposition of samples under inert atmosphere. The results are shown in Figure 7. In the TGA curves (Figure 7a), the onset decomposition temperature of RDX and RDX@PVDF microspheres (3/1) were 205 °C and 214 °C (Figure 7a), respectively, indicating that the addition of PVDF can raise the onset decomposition temperature of RDX. Moreover, the DSC curves of pure RDX and RDX@PVDF had an endothermic peak at approximately 205 °C, which represented the melting peak of RDX. An exothermic peak followed, with a peak temperature at approximately 230 °C, indicating the decomposition of RDX. The exothermic peak temperature of pure RDX was 232 °C (Figure 7b), and those of the RDX@PVDF microspheres were approximately 242 °C~244 °C. In the DSC curve of PVDF (Figure 7c), 161 °C was the melting peak of PVDF, which can also be seen in the DSC curve of the RDX@PVDF sample (Figure 7a). However, the decomposition endothermic peak of PVDF at 479 °C could not be observed, because the temperature of the RDX@PVDF sample was examined in the range of 30 °C to 300 °C.

Combined with the data obtained by the TGA curves, the initial decomposition temperature and the decomposition peak temperature of the products passivated with PVDF were significantly increased, indicating that the coating of PVDF can indeed improve the thermal stability of RDX. Furthermore, as the PVDF content increased, the thermal stability of RDX@PVDF microspheres became better, and when the added mass loading of PVDF was one-third of RDX, the samples possessed optimum thermal stability with an onset temperature of 214 °C and a peak temperature of 244 °C. Considering that the exorbitant contents of PVDF reduce particle energy, the RDX@PVDF microspheres with higher PVDF content were not tested.

To compare the thermal stability of the RDX@NC microspheres with that of the RDX@PVDF microspheres, the RDX@NC microspheres and the RDX@PVDF microspheres were measured by TGA-DSC, and the results are shown in Figure 8. The onset temperature of RDX@NC was 180 °C (Figure 8a), which was greatly reduced when compared with that of the RDX and RDX@PVDF microspheres (Figure 7a). In addition, two exothermic peaks of RDX@NC DSC curves are presented in Figure 8b: the first exothermic peak was at 217 °C and the second one was at 242 °C, which represent the decomposition of NC and RDX, respectively. The decomposition of NC started at approximately 180 °C [34] and the first peak temperature of 217 °C of RDX@NC microspheres was much lower than that of the RDX (232 °C) and RDX@PVDF microspheres (242 °C~244 °C). The results showed that the property of thermal stability of the RDX@NC microspheres decreased with the coating of NC, due to the high thermal sensitivity of NC, which further showed that PVDF is more suitable for coating RDX to improve thermal stability.

### 3.5. Impact and Friction Sensitivity of Samples

The friction sensitivity and the impact sensitivity of the samples were tested by the explosion probability method, and the results are shown in the Table 2 and Table 3. The results for friction sensitivity show that the explosion probability of RDX was 73% at 360 N and 52% at 324 N. The sensitivity of RDX coated by NC increased with the explosion probability of 100% at both 324 N and 360 N. However, the sensitivity of particles coated by PVDF decreased, and the particles were not even initiated at 360 N, which was the largest loading force of the instrument.

The impact results showed that when the drop hammer was 2 kg and the drop height was 30 cm, the explosion probability of RDX and RDX@NC were both 100%. After coating with PVDF, the impact sensitivity of the microspheres decreased significantly; the explosion probability decreased to 38% (RDX/PVDF = 5/1), 32% (RDX/PVDF = 4/1), and 21% (RDX/PVDF = 3/1), respectively. According to the test results of the friction sensitivity and the impact sensitivity, the mechanical sensitivity of RDX passivated with PVDF-employed electrospray technology was significantly reduced, which was attributed to the spheroidization of the particles [35], the reduction in the particle size, and the protection of the coating layer.

## 4. Conclusions

In conclusion, in this study we explored an efficient and facile approach, electrospray assembly technology and introduced PVDF to passivate RDX, constructing ingenious core-shell structured RDX@PVDF microspheres without changing the crystal form of RDX. PVDF content was a key parameter in adjusting the thermal stability, friction sensitivity, and impact sensitivity of the resulting microspheres. When the ratio of RDX:PVDF reached 3:1, the onset decomposition temperature increased by 9 °C; the impact explosion probability decreased by 79% when the drop hammer was 2 kg and the drop height was 30 cm; and the explosion probability also reduced from 73% to 0 under the 360 N friction force. Therefore, coating with PVDF on RDX through electrospray assembly technology is expected to be applied in improving the prediction of environmental adaptability for energetic materials.

## Figures and Tables

**Figure 1 polymers-14-04262-f001:**
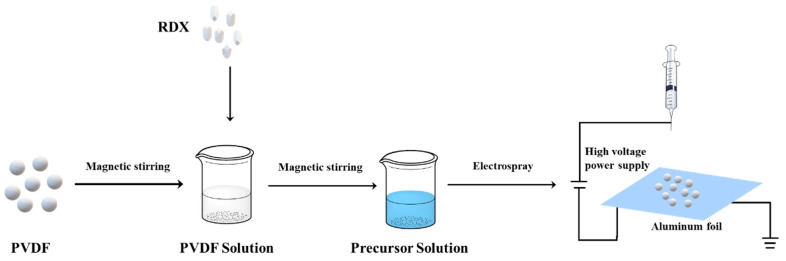
Scheme of preparation of RDX@PVDF microspheres employed by electrospray technology.

**Figure 2 polymers-14-04262-f002:**
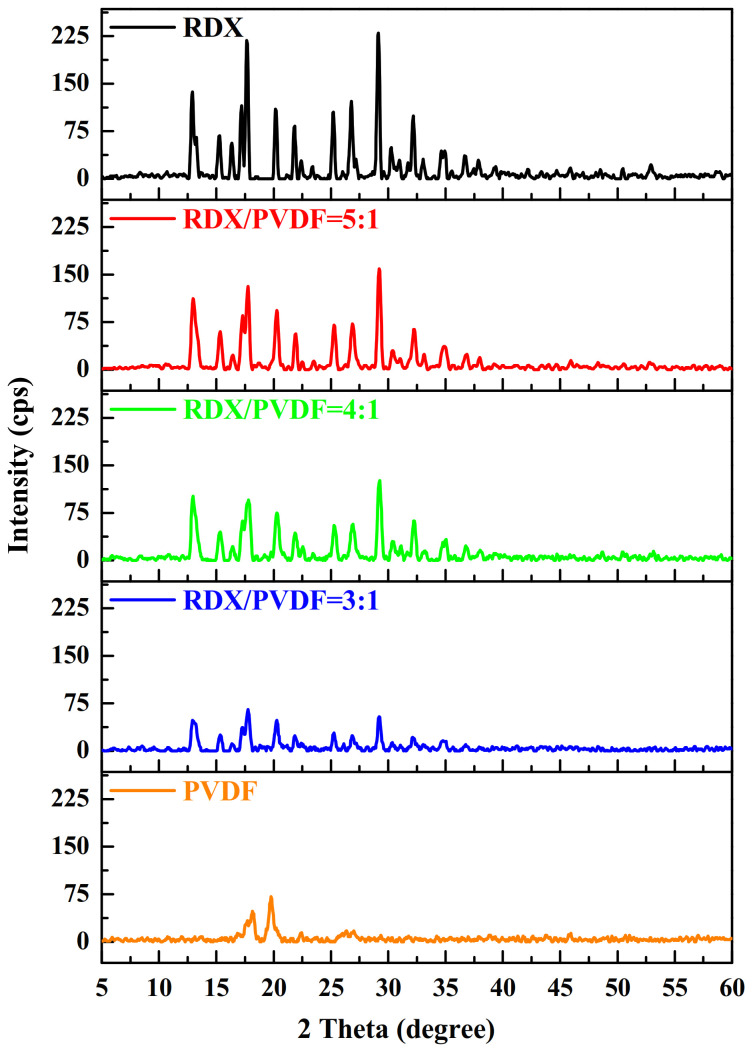
X-ray Diffraction patterns of raw RDX and RDX@PVDF microspheres.

**Figure 3 polymers-14-04262-f003:**
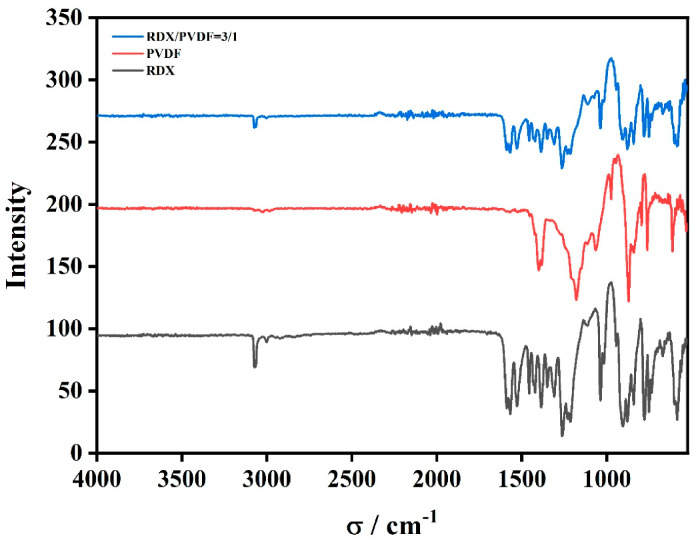
FTIR spectra of raw RDX and RDX@PVDF microspheres.

**Figure 4 polymers-14-04262-f004:**
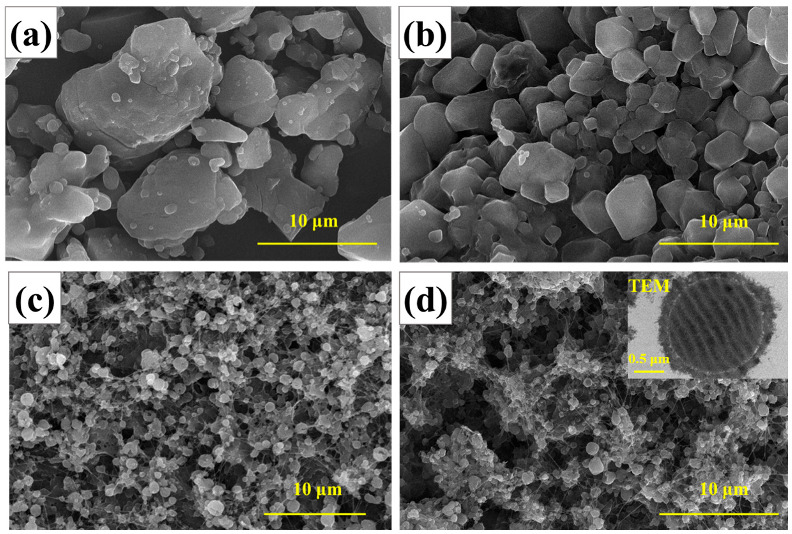
SEM images of raw RDX and RDX@PVDF microspheres: (**a**) raw RDX; (**b**) RDX@PVDF (5:1); (**c**) RDX@PVDF (4:1); and (**d**) RDX@PVDF (3:1).

**Figure 5 polymers-14-04262-f005:**
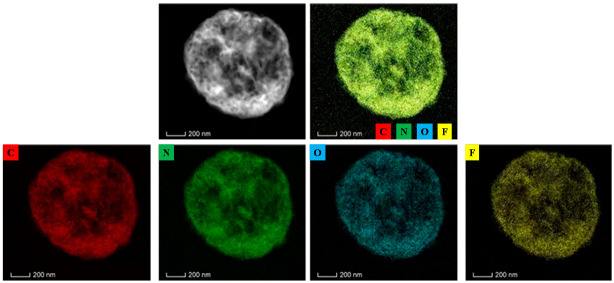
Energy dispersive X-ray spectroscopy mapping of RDX@PVDF particles (RDX/PVDF = 3/1).

**Figure 6 polymers-14-04262-f006:**
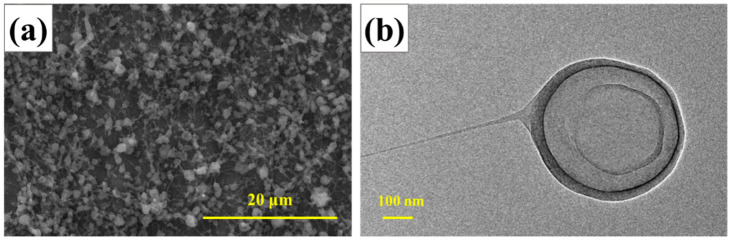
The images of RDX@NC (3:1) Microspheres: (**a**) SEM image; (**b**) TEM image.

**Figure 7 polymers-14-04262-f007:**
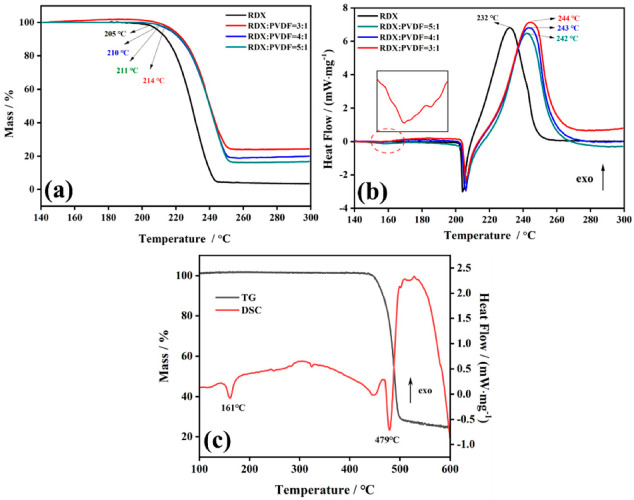
TGA/DSC of RDX@PVDF microspheres at 10 K/min: (**a**) TG curves of raw RDX and RDX@PVDF; (**b**) DSC curves of raw RDX and RDX@PVDF; and (**c**) TG-DSC curves of PVDF.

**Figure 8 polymers-14-04262-f008:**
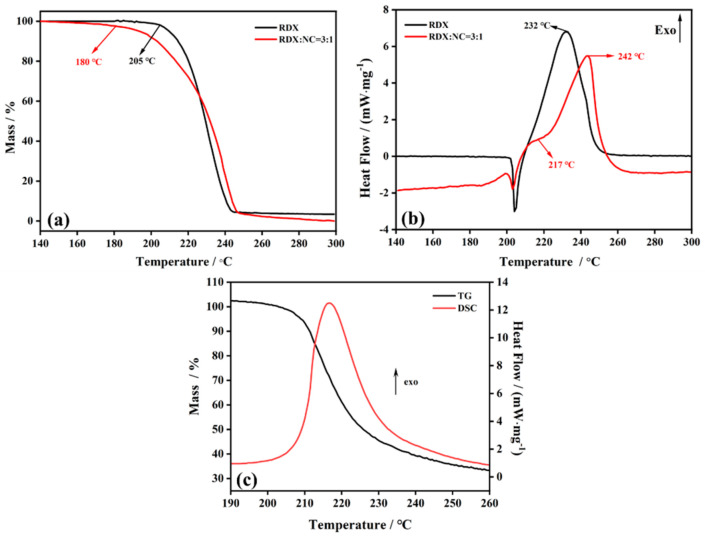
TGA/DSC of RDX@NC microspheres at 10 K/min: (**a**) TG curves of raw RDX and RDX@NC; (**b**) DSC curves of raw RDX and RDX@NC; and (**c**) TG-DSC curves of NC.

**Table 1 polymers-14-04262-t001:** The sample formulae of the RDX/PVDF precursor solution.

Samples	Mass Ratio of RDX/PVDF	RDX (mg)	PVDF (mg)	Acetone (mL)	DMF (mL)
1	3:1	195	65	6	2
2	4:1	208	52	6	2
3	5:1	217	43	6	2

**Table 2 polymers-14-04262-t002:** The friction sensitivity results of samples.

Samples	Loading Force/N	Explosion Probability
RDX	360	73%
	324	52%
RDX@NC	360	100%
	324	100%
RDX@PVDF (5:1)	360	0
RDX@PVDF (4:1)	360	0
RDX@PVDF (3:1)	360	0

**Table 3 polymers-14-04262-t003:** The impact sensitivity results of samples.

Samples	Explosion Probability
RDX	100%
RDX@NC	100%
RDX@PVDF (5:1)	38%
RDX@PVDF (4:1)	32%
RDX@PVDF (3:1)	21%

## Data Availability

Not applicable.

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
