# Peer review of "Preparation of Core-Shell-Structured RDX@PVDF Microspheres with Improved Thermal Stability and Decreased Mechanical Sensitivity"

_polymers, 2022, doi:10.3390/polym14204262_

Round 1
Reviewer 1 Report
In this manuscript, the authors presented the core-shell structured RDX@PVDF microspheres, prepared by the electrospray method, to reduce the RDX sensitivity. The prepared microspheres were characterised by thermogravimetric analysis (TGA), differential scanning calorimetry (DSC), X-ray diffraction (XRD), scanning electron microscope (SEM), and transmission electron microscope (TEM). The sensitivity of the microspheres was also investigated in an impact and friction test. The experiments are sound and the results are interesting. However, more discussions need to be provided to highlight the novelty and significance of the work. Some points to consider are below.
1. In the introduction, the authors should further discuss the motivation of this work (i.e., to reduce the sensitivity of RDX) and the advantages of the electrospray method compared to other methods.
2. The authors mentioned the electrospray method can control the particle size and surface morphology. How were the particle size and surface morphology controlled in this work? What parameters should be considered? For instance, nozzle size and shape, flow rate, and distance between the nozzle and the receiver are some parameters that were mentioned in the experiment section.
3. For the impact test, a hammer weight of 2kg was dropped at the height of 30 cm. It may be useful to provide additional data at different weights and heights. Similarly, the friction test was performed with only two loading forces of 324 N and 360 N. Is there any specific reason to choose these values? If the authors have more data, the results can be presented as graphs instead of tables, which is more attractive to readers.
Author Response
- In the introduction, the authors should further discuss the motivation of this work (i.e., to reduce the sensitivity of RDX) and the advantages of the electrospray method compared to other methods.
Response: We thank the Reviewer for this positive comment. These two points are the focus of our paper and have been explained in the introduction.
The coating RDX was emphasized to reduce its sensitivity in the introduction. Recently, the main coating materials employed to reduce sensitivity of RDX include Viton (dipolymers of hexafluoropropylene and vinylidene fluoride) [9], DAAF (3,3’-diamino-4,4’-azoxyfurazan) [10], F2604 [11], PMMA (polymethyl methacrylate) [12], MUF (melamine-urea-formaldehyde) [13] and NC [14-17]. Xinlei Jia [14] fabricated honeycomb-like-structured nanoparticles in which partial RDX crystals are encapsulated in PMMA-PVA (polymethylmethacrylate-polyvinyl acetate) coating, manifested that the addition of PMMA could effectively improve the thermal stability of RDX. Jian Yao [11] successfully prepared RDX/F2604 composite particles by electrospray assembly technology, which possess lower impact sensitivity and friction sensitivity compared with that of raw RDX.
The advantages of electrospray were also explained: electrostatic spray technology has been widely used in the preparation of composite particles [31-32], because it can easily realize the regulation of particle size from tens of nanometers to tens of microns, but also could control the surface morphology of composite particles [33].
- The authors mentioned the electrospray method can control the particle size and surface morphology. How were the particle size and surface morphology controlled in this work? What parameters should be considered? For instance, nozzle size and shape, flow rate, and distance between the nozzle and the receiver are some parameters that were mentioned in the experiment section.
Response: We thank the Reviewer for this positive comment.
The size and surface morphology of the sample are usually achieved by adjusting the parameters of the electrospray method. Corresponding parameters are as follows: the voltage is 23kV; the flow rate is 1.5 ml∙h-1; distance between nozzle and receiver is 8 cm; nozzle size is inner diameter 0.51 mm and outer diameter 0.80 mm.
- For the impact test, a hammer weight of 2kg was dropped at the height of 30 cm. It may be useful to provide additional data at different weights and heights. Similarly, the friction test was performed with only two loading forces of 324 N and 360 N. Is there any specific reason to choose these values? If the authors have more data, the results can be presented as graphs instead of tables, which is more attractive to readers.
Response: We thank the Reviewer for this insightful comment. The tests of impact and friction sensitivity were examined according to 《GJB 772A-97 Explosive test method》.

Reviewer 2 Report
1) It is not entirely clear for which specific application the authors plan to use the obtained core/shell structures;
2) It was desirable to present IR spectra for RDX, polyphenylidene fluoride and shell structures obtained on its basis;
3) The presence of a shell of polyvinylidene fluoride will lead to a change in the density of RDX. The true density of such shell structures has been determined?
4) The effect of the PVDF jacket on a number of important parameters such as detonation velocity and enthalpy has been evaluated?
5) In Figure 6, for a better perception of the results, it is necessary to present a TGA and DSC thermogram for polyvinylidene fluoride, and in Figure 7 - for cellulose nitrate;
Author Response
1) It is not entirely clear for which specific application the authors plan to use the obtained core/shell structures;
Response: We thank the Reviewer for this positive comment. Hexahydro-1,3,5-trinitro-1,3,5-triazine (RDX), a high energy nitroamine explosive with strong explosive power and high energy density, has been widely used in casting explosives and high energy propellants. However, due to its high sensitivity, it is prone to safety accidents in the process of use, transportation and storage, endangering the safety of personnel and property. Therefore, it is necessary to reduce its sensitivity. The sensitivity of RDX was reduced by coating the RDX surface, thereby reducing the possibility of safety accidents, which is the ultimate use of the core-shell structure.
2) It was desirable to present IR spectra for RDX, polyphenylidene fluoride and shell structures obtained on its basis;
Figure 1 IR spectra of RDX/PVDF
Response: We thank the Reviewer for this positive comment. The IR spectra of the samples has been tested, and the results are shown in Figure 1. There are no chemical reactions between RDX and PVDF after coating, but only simple physical coating. The result is consistent with XRD. Whether RDX was successfully coated by PVDF could not be determined from IR spectra. However, the results of TEM and EDS showed that PVDF was successfully coated on the surface of RDX.
3) The presence of a shell of polyvinylidene fluoride will lead to a change in the density of RDX. The true density of such shell structures has been determined?
Response: We thank the Reviewer for this positive comment. Due to the low yield of the obtained samples, the density of the samples was not measured. In this paper, the research focus of this paper is to reduce the sensitivity of RDX, and the measurement of density will be our next work.
4) The effect of the PVDF jacket on a number of important parameters such as detonation velocity and enthalpy has been evaluated?
Response: We thank the Reviewer for this insightful comment. The focus of this paper is to reduce the sensitivity of RDX, and the corresponding parameters of PVDF are not evaluated, which will be considered in our next work.
5) In Figure 6, for a better perception of the results, it is necessary to present a TGA and DSC thermogram for polyvinylidene fluoride, and in Figure 7 - for cellulose nitrate;
Response: We thank the Reviewer for this positive comment. The TG-DSC curves of PVDF and NC have been added to Figure 6 and Figure 7 (The TG-DSC pictures of PVDF and NC are in the uploaded response). The conditions are as below: under a high-purity argon environment with a flow rate of 20 mL∙min-1, and the heating rate is 10 °C ∙min-1.
The TG-DSC curve of PVDF was shown in Figure 2. PVDF began to decompose at 473 ℃ and lost nearly 80% wt. 161 ℃ is the melting peak of PVDF, and 479 ℃ is the decomposition exothermic peak of PVDF. Since the temperature range of RDX/PVDF samples was 30-300 ℃, 479 ℃ of the decomposition exothermic peak was not observed. However, the melting peak at about 161 ℃ can be observed in the DSC curve of RDX/PVDF. The result was depicted in Figure 3:
The DSC result of nitrocellulose was shown in Figure 4, and the exothermic peak temperature was 216 ℃, which was consistent with the results of RDX/NC.

Reviewer 3 Report
The provided paper is well written using good English, while the study is logically and physicochemically correctly performed and has no flaws. I guess the manuscript can be accepted for publishing in Polymers after the authors do minor revisions, which I noted in the file because there is no line numeration in the paper.

Author Response
The provided paper is well written using good English, while the study is logically and physicochemically correctly performed and has no flaws. I guess the manuscript can be accepted for publishing in Polymers after the authors do minor revisions, which I noted in the file because there is no line numeration in the paper.
Response: We thank the Reviewer for this positive comments. These detailed comments have been modified with blue font. The uploaded Word document is the new version.

Round 2
Reviewer 1 Report
The authors answered all the questions. The manuscript can be accepted for publication.
Author Response
Response: We thank the Reviewer for these positive comments. The content of research has been modified, and the modified content has been uploaded in the attachment.

Reviewer 2 Report
1) IR spectra must be included in the text of the article or additional information describing the main bands characteristic of RDX, polyvinylidene fluoride, and the shell structure based on them.
2) For figures 6 and 7, give the interpretation of the figures under the letters a), b), c).
3) Description of the TGA/DSC curve for polyvinylidene should be included in the text of the article.
4) It is necessary to add a figure showing the presence of a melting peak for the shell structure (Figure 3 in the answer to the reviewer).
5) On the given TGA/DSC curve for polyvinylidene fluoride, 479°C is the temperature of the endothermic peak, and the exothermic peak is observed in the temperature range of 500-540°C.
Author Response
1) IR spectra must be included in the text of the article or additional information describing the main bands characteristic of RDX, polyvinylidene fluoride, and the shell structure based on them.
Response: We thank the Reviewer for this positive comment. The FTIR absorption spectra of raw RDX, PVDF and RDX@PVDF (3/1) was depicted in Figure 3. About raw RDX (black curve), the absorption peak at 3074 cm-1 is ascribable to the ‐CH stretching vibration; the three absorption peaks at about 1569, 1528, and 1386 cm-1 corresponds to the stretching vibrations of ‐NO2; another peak at 1264 cm-1 is attributed to the nitramine; the peak at 905 cm-1 originates from the stretching vibration of RDX ring. Compared with pure RDX, the peak of IR spectra of RDX@PVDF has little change significantly, and it is difficult to judge the relevant support of its structure. Therefore, the SEM, TEM and EDS were used to further illustrate the core-shell structure.
2) For figures 6 and 7, give the interpretation of the figures under the letters a), b), c).
Response: We thank the Reviewer for this positive comment. The interpretation of the figures under the letters a), b), c) has been added.
Fig. 6 TGA/DSC of RDX@PVDF microspheres at 10 K/min: (a) TG curves of raw RDX and RDX@PVDF; (b) DSC curves of raw RDX and RDX@PVDF; (c) TG-DSC curves of PVDF
Fig. 7 TGA/DSC of RDX@NC microspheres at 10 K/min: (a) TG curves of raw RDX and RDX@NC; (b) DSC curves of raw RDX and RDX@NC; (c) TG-DSC curves of NC
3) Description of the TGA/DSC curve for polyvinylidene should be included in the text of the article.
Response: We thank the Reviewer for this positive comment. In the DSC curve of PVDF (Figure 7c), 161℃ is the melting peak of PVDF, which can also be seen in the DSC curve of RDX@PVDF sample (Fig. 6a). However, the decomposition endothermic peak of PVDF is at 479 °C cannot be observed, because the temperature of RDX@PVDF sample was examined in the range of 30-300 °C.
4) It is necessary to add a figure showing the presence of a melting peak for the shell structure (Figure 3 in the answer to the reviewer).
Response: We thank the Reviewer for this positive comment. The presence of a melting peak for the shell structure has been added to the Fig. 6.
5) On the given TGA/DSC curve for polyvinylidene fluoride, 479°C is the temperature of the endothermic peak, and the exothermic peak is observed in the temperature range of 500-540°C.
Response: We thank the Reviewer for this positive comment. 479 °C is the temperature of the endothermic peak, not an exothermic peak. That’s the kind of mistake we described, which should not exist. We will be careful in the next step.
